# Sensitivity of the Fasciae to the Endocannabinoid System: Production of Hyaluronan-Rich Vesicles and Potential Peripheral Effects of Cannabinoids in Fascial Tissue

**DOI:** 10.3390/ijms21082936

**Published:** 2020-04-22

**Authors:** Caterina Fede, Carmelo Pirri, Lucia Petrelli, Diego Guidolin, Chenglei Fan, Raffaele De Caro, Carla Stecco

**Affiliations:** Department of Neurosciences, Institute of Human Anatomy, University of Padova, Via A. Gabelli 65, 35121 Padova, Italy; caterina.fede@unipd.it (C.F.); carmelop87@hotmail.it (C.P.); lucia.petrelli@unipd.it (L.P.); diego.guidolin@unipd.it (D.G.); chenglei.fan@studenti.unipd.it (C.F.); rdecaro@unipd.it (R.D.C.)

**Keywords:** fascia, hyaluronan, vesicles, endocannabinoid, agonist, antagonist, inflammation, fibrosis, pain, peripheral effect

## Abstract

The demonstrated expression of endocannabinoid receptors in myofascial tissue suggested the role of fascia as a source and modulator of pain. Fibroblasts can modulate the production of the various components of the extracellular matrix, according to type of stimuli: physical, mechanical, hormonal, and pharmacological. In this work, fascial fibroblasts were isolated from small samples of human fascia lata of the thigh, collected from three volunteer patients (two men, one woman) during orthopedic surgery. This text demonstrates for the first time that the agonist of cannabinoid receptor 2, HU-308, can lead to in vitro production of hyaluronan-rich vesicles only 3–4 h after treatment, being rapidly released into the extracellular environment. We demonstrated that these vesicles are rich in hyaluronan after Alcian blue and Toluidine blue stainings, immunocytochemistry, and transmission electron microscopy. In addition, incubation with the antagonist AM630 blocked vesicles production by cells, confirming that release of hyaluronan is a cannabinoid-mediated effect. These results may show how fascial cells respond to the endocannabinoid system by regulating and remodeling the formation of the extracellular matrix. This is a first step in our understanding of how therapeutic applications of cannabinoids to treat pain may also have a peripheral effect, altering the biosynthesis of the extracellular matrix in fasciae and, consequently, remodeling the tissue and its properties.

## 1. Introduction

Myofascial pain is a common clinical disorder, beginning as acute pain within the musculoskeletal system with a referred autonomic phenomenon, which includes pain, decreased ranged of motion, and weakness [1]. Myofascial pain syndromes (MPSs) can be categorized as primary when unrelated to other disorders or secondary when associated with comorbid medical conditions [2]. Sustained contractile activity of the trigger points, identified as discrete foci of hypercontracted areas within a muscle, causes local ischemia, hypoxia, and neurophysiologic modifications at nociceptors, causing pain sensitization and deep referred pain, distant from the initial stimulus. For many years, the focus of myofascial pain was believed to involve only muscle, but the role of the fascia has recently been reconsidered [3]. However, exactly how the fasciae can cause pain and how they may respond to various drug therapies is still not clear.

Myofascial pain affects as many as 85% of people at least once during their lifetime [4], with prevalence ranging from 21% of general orthopedic patients to 93% of patients in specialized pain centers [5]. MPS is an important and often overlooked cause of disability in clinical practice. Both men and women are affected, with a higher risk in middle-aged sedentary women. Regular and vigorous activity, such as moderate-intensity daily exercise, seems to be protective [1,6]. The recent literature demonstrates that myofascial pain effectively responds to therapy with cannabinoids, which can provide analgesic relief for chronic pain disorders [7,8]. In the United States over the last decade, there has been increasing availability and use of cannabis for medical purposes and it has become important to provide the public with accurate information as to the use and effectiveness of painkillers [9,10,11]. For this reason, having already demonstrated in one of our previous works that the human fasciae express both endocannabinoid receptors CB_1_ and CB_2_ [12], we stimulated fascial cells with an agonist of CB_2_ receptors, to better understand the effects of these molecules in peripheral tissue; we then confirmed the CB_2_-mediated effect blocking the action with a specific antagonist. Endocannabinoids are endogenous lipid mediators with a wide range of biological effects similar to those of marijuana, and can be generated in several cell types, e.g., both in the brain and in various peripheral tissues [13,14,15]. Endocannabinoids exert their biological effects via two main G-protein-coupled cannabinoid receptors, CB_1_ and CB_2_ [16,17]. Their presence in human fascial tissue constitutes a first step to know more about the role of the fasciae as pain generators and the efficacy of some fascial treatments [12]. We believe that the present work may improve our understanding of the effects of stimulation with a synthetic cannabinoid in peripheral tissue.

## 2. Results

The incubation of isolated fibroblasts with a small percentage of DMSO (0.5%) did not affect cell viability (Figure 1), and neither cell growth in the in vitro culture after cell attachment (Figure 2A). Density, morphology, stains, and viability remained the same with respect to cells maintained in culture without DMSO incubation. After these preliminary verifications, we chose to carry out the whole analysis using samples to which we had added 0.5% DMSO, as control cells, to better verify any change caused only by incubation with the synthetic cannabinoid HU-308. All results described here were the same in all three cell lines.

The dose-response curve demonstrated that the dose 1 µM was not cytotoxic for either HU-308 or AM630 (89.2 ± 4.3% and 73.4 ± 7.8% of viability, respectively, Figure 1), but it was not enough to produce visible effects in the cells. Instead, doses of 4 and 5 µM were toxic for fascial cells, with a statistically significant decrease in viability (Dunnett’s test, *p* < 0.05) with respect to control cells. Viability decreased to 63.0 ± 4.1% and 54.1 ± 0.8%, respectively, after incubation with HU-308 4 and 5 µM, and to 33.7 ± 6.2% and 27.2 ± 2.9% with AM630 (Figure 1). Therefore, the best non-toxic dose for cells, and one capable of producing visible effects in cell culture, was 2.5 µM: cell viability remained at 87.4 ± 2.9% after incubation with HU-308, with no statistically significant differences with respect to control cells. The dose of AM630 equal to 2.5 µM caused a reduction in viability (70.5 ± 8.5%), but a dose at least equal to that of the agonist was needed to counteract its effect. In any case, the reduction in viability was not significantly different (*p* > 0.05) from the control cells. For this reason, the treatment with agonist and/or antagonist was conducted at the dose 2.5 µM.

After incubation with the agonist of CB_2_ receptors, after only 1 hour of treatment, the formation of cytoplasmic vesicles (Figure 2B) has been noticed. The vesicles were visible in the cytoplasm of cells even after 2.5 h (Figure 2C,D) and especially after 4 h, near the nucleus and in cellular extensions (Figure 2E,F). After 6 h of incubation, the detection of vesicles visible in the cells decreased, probably because their content had already been exocytosed in the extracellular ambient (Figure 2G,H). Therefore, the timing of 4 h was decided to fix the cells. In addition, at this timing no variations in cells density were noted, although some cells, especially in non-confluence areas of the culture, had started to change their morphology, appearing with long extensions rich in vesicles (Figure 2F).

After fixation by adding the fixative solution directly into the well with the medium, it was possible to avoid washing away any vesicles in the cellular excretion phase. Some of them appeared in the excretion phase of the cells, others already exocytosed were lost in the fixing solution. In any case, no changes in cell density were noted after treatment.

Staining confirmed the presence of vesicles in the treated cells with respect to control cells (Figure 3). The vesicles showed intense coloring, both with Toluidine blue (Figure 3B) and Alcian blue (Figure 3D), demonstrating the presence of mucopolysaccharides in the vesicles. Figure 3D also clearly shows some vesicles close to the plasma membrane, larger and less intensely stained (probably residues of already exocytosed vesicles, which are also visible in Figure 2H). Control cells did not show any vesicles with either staining solution (Figure 3A–C).

The analysis of semithin sections confirmed the presence of material inside the cytoplasm of the treated cells and confirmed the presence of vesicles (Figure 4D–F). The nuclei appeared undamaged with respect to control cells. The latter showed no production of vesicles in any cell (Figure 4A–C). TEM analysis (Figure 5) also confirmed the presence of vesicles in the treated samples, whereas cells treated only with DMSO without HU-308 appeared with no morphological changes (Figure 5A). Cells treated with the CB_2_ agonist revealed vesicles near the Golgi apparatus (Figure 5B), as well as in the cytoplasmic extensions of treated cells (Figure 5C–E). In general, the mean dimensions of the vesicles are on the order of 0.5–1 to a maximum of 10 microns. The details of the produced vesicles clearly show that they contained amorphous material secreted by the cells (Figure 5C–F). The nuclear membrane was not damaged by the treatment, neither in the samples (control with DMSO) nor in cells treated with DMSO + HU-308.

The immunocytochemistry analysis with hyaluronic acid binding protein (HABP) confirmed that the vesicles were rich in hyaluronan (HA). Figure 6D–F shows that all the vesicles produced by the cells were very positive to HABP. Instead the control cells (Figure 6B,C) showed neither cytoplasmic extensions nor positive vesicles; there was less positivity to HABP, which was homogenous in the cytoplasm. The negative control (Figure 6A) confirmed the specificity of the staining: the cells were not positive and only the nucleus was colored after incubation with hematoxylin.

Treatment with the CB_2_ antagonist confirmed that the production of HA-rich vesicles was mediated by the CB_2_ receptor. In fact, no modifications in the cells were visible after incubation with AM630 by either bright field microscopy analysis (Figure 7B) or HABP staining (Figure 7E). In addition, the vesicles produced by HU-308 (Figure 7A–D) were no longer visible after co-administration of the antagonist (treatment with HU-308 and AM630, Figure 7C–F), confirming that the release of hyaluronan was a cannabinoid-mediated effect.

## 3. Discussion

This work demonstrates that a synthetic cannabinoid can lead to the production of hyaluronan and HA-rich vesicles in only a few hours in an in vitro culture of fascial fibroblasts. We noted no differences due to the gender or age of the patient from whom we isolated the fascial fibroblasts: after only 1 h the cells seeded in the multiwells and treated with HU-308 2.5 µM started to produce vesicles, and after 4 h, these vesicles rich in hyaluronan were visible in both cytoplasm and extracellular space.

It is widely recognized that extracellular vesicles constitute a fundamental signal transmission in the central nervous system. Venturini and coauthors recently demonstrated that astrocytes are able to convey messages and transfer neuroglobin through extracellular vesicles, which selectively target neurons [18]. However, it is widely recognized that different cell types can release micro-vesicles, which operate as mediator of the inter-cellular communication ensuring short- and long-range exchange of information [19,20]. Guescini et al. demonstrated that also skeletal muscle cells release small vesicles of 50–80 nm in diameter: they carry mitochondrial DNA (mtDNA) and signaling proteins, and regulate the communication processes within skeletal muscles and between skeletal muscles and other organs [19]. In general, micro-vesicles are circular membrane fragments that maintain the characteristics of the cell of origin and contain cytosol. They differ in size and molecular composition, and can be distinguished in exosomes with endosomal origin (from 30 to 120 nm), or shedding vesicles (from 100 nm to 1 μm), rich in phosphatidylserine and proteins associated with membrane lipid rafts [20]. Besides, we have to distinguish the micro-vesicles secreted in a constitutive way from the ones secreted after a specific cell activation, such as the vesicles described in this work. The latter are on the on the order of 0.5–1 to 10 microns, and they are rich in hyaluronan and glycosaminoglycans. A deep analysis of biochemical characteristics and possible functions of the material contained in the HA-rich vesicles is a task for future investigations. However, this work demonstrated that fascial cells respond to the endocannabinoid system, regulating and remodeling the formation of the extracellular matrix (ECM), as already demonstrated for other types of stimuli, such as changes in hormonal levels [21]. The increase of the secreted hyaluronan can cause a greater, at least temporary, fluidity of the tissue, thanks to the role of the HA in facilitating the gliding between the fascial layers within and underlying the deep fascia during movement [22].

One limitation of our study is that it only shows qualitative images of the treated cells, with no quantification of the production of the vesicles, in either intracellular space or the extracellular medium. For this type of evaluation, it is necessary to consider the amount of the vesicles inside the cells and released in the culture medium during time, aspect that needs to be deepened. However, this work really constitutes a first step in our understanding of the peripheral effect of cannabinoids in fascial cells, and also demonstrates for the first time the mechanism of HA production after a stimulus and the inhibition by an opposite stimulus. In fact, after co-administration of a specific antagonist of CB_2_, the production of vesicles and exocytosis of HA by cells was blocked, confirming that these effects were mediated by a specific mechanism induced by the linkage of HU-308 to the CB_2_ receptors of the fascial cells.

It has already been widely demonstrated that hyaluronan is a critical element for ECM composition and remodeling [23,24]. It is a simple linear polymer, immensely hydrophilic, which confers large volumes of hydration and contributes to the turgor and flexibility of a tissue [25]. In the deep fascia it affects the movement of HA-containing fluid layers within and underlying the deep fascia, permitting smooth gliding between these structures during movement, and consequently improving the tissue adaptability [22]. Changes in the viscosity of HA-containing fluids influence the behavior of the tissue. In addition, HA is considered one of the key players in the tissue regeneration process [26]. Arasu et al., 2017, demonstrated that human mesenchymal stem cells can secrete HA-coated vesicles, contributing to the mechanisms of HA-mediated tissue regeneration. After incubating the tissues with lipopolysaccharide, the authors demonstrated induced filopodial growth, plasma membrane bleeding, and HA secretion, which can contribute to ECM remodeling, directly or indirectly by interacting with ECM-producing cells [27]. The production of hyaluronan induced by cannabinoids can increase the ability of the collagen bundles inside the fasciae to glide with respect to others, and consequently to improve tissue adaptability. It is confirmed, in fact, that changes in fluid content, crosslinks, and molecular organization, and the contents of specific ECM molecules, can modify the mechanical properties and stiffness or elasticity of fascial tissues [28]. By increasing the adaptability and hydration of the tissue, the nervous fibers widespread in the fascia may be less stimulated [29].

Another possible effect of the endocannabinoid system stimulation in fascial tissues may be due to suppression of pro-inflammatory cytokines, such as Interleukin-1beta (IL-1beta) and Tumor necrosis factor (TNF-alpha), and to the increase of anti-inflammatory cytokines, which are known to be able to provide anti-fibrotic activity and relief of myofascial pain [30,31].

In conclusion, this work can constitute a demonstration that the myofascial pain can effectively respond to cannabinoids, which are able to provide analgesic relief for chronic pain disorders. The cannabinoids and their receptors on the fascial fibroblast cause a rapid production of HA, demonstrating for the first time that the cannabinoids may have an effect not only on pain perception, due to central nervous system phenomena, but also direct peripheral effects, which result in modification of the structure of fascial tissue.

## 4. Materials and Methods

### 4.1. Preparation of HU-308 and AM630 Solutions

HU-308(4-[4-(1,1-Dimethylheptyl)-2,6-dimethoxyphenyl]-6,6-dimethylbicyclo[3.1.1]hept-2-ene-2-methanol) was purchased as a potent and selective CB_2_ agonist by Tocris Bioscience (Bio Techne s.r.l., Milano, Italy). According to indications, 1 mg powder of HU-308 was diluted in 2.41 mL DMSO (dimethyl sulfoxide) to prepare a 5 mM stock solution, maintained at −20 °C before use.

AM630(6-Iodo-2-methyl-1-[2-(4-morpholinyl)ethyl]-1H-indol-3-yl](4-methoxyphenyl) methanone) was purchased as a selective CB_2_ cannabinoid antagonist by Merk Life Science S.r.l. In, Milan, Italy. Moreover, 5 mg of powder were diluted in 1 mL of DMSO to obtain 10 mM stock solution, maintained at 4 °C before use.

### 4.2. Cell Isolation and Culture

Cells were isolated from human fascia lata of the thigh (~1cm × 1 cm) collected from three volunteer patients: two men and one woman, average age 73 ± 5, who were undergoing elective surgical procedures at the Orthopedic Clinic of the University of Padova. The ethical regulations regarding research on human tissues were carefully followed (approval no. 3722/AO/16, study approved on 21 April 2016 by the Ethical Committee for clinical trials in the province of Padova) and written informed consent was obtained from each donor. Samples were transported to the laboratory in phosphate buffered saline (PBS) containing 1% penicillin and streptomycin, within a few hours of collection. Fasciae were digested with Collagenase B 0.1% in Hank’s Balanced Salt Solution (HBSS) overnight, centrifuged at 480 g for 5 min, and transferred to tissue flasks with DMEM 1 g/L glucose, 10% FBS, and 1% penicillin–streptomycin antibiotic. Isolated fibroblasts were characterized by immunohistochemical staining with anti-Fibroblast Surface Protein (1B10) antibody (1:100, Mouse monoclonal antibodies, AbCam Cambridge, UK), as previously described in one of our previous works [12]. Cell culture were maintained at 37 °C, 95% humidity and 5% CO_2_, and used from passage 3rd to 9th.

### 4.3. Cell Treatment with CB2R Agonist and/or Antagonist

Isolated cells were plated (150 cells/mm^2^ in 500 μL in 24-multiwells) and allowed to attach for 48 h at 37 °C. To evaluate HU-308 and AM630 cytotoxicity, the two substances were diluted to appropriate concentrations selected to evaluated dose/survival (0-1-2.5–4-5 µM) in cell culture medium. The HU-308 and AM630 stock solutions were firstly diluted 1:10 in DMSO; then all the treatments solutions were prepared in DMEM by serial dilution with the final volume of DMSO of only 2.5 µL: in this way, the concentration of DMSO in the final solution was reduced to a minimum and consequently also the toxicity of the solvent. In any case, in each experiment, one control sample was treated with 2.5 µL DMSO in 500 µL (0.5%) but did not receive HU-308/AM630. Treatment solutions were immediately administered to cells.

### 4.4. Dose-Response and Time-Course Analysis

To evaluate the time response of fascial cells to the treatment, fibroblasts were treated with HU-308 2.5 µM for 1, 2.5, 4, and 6 h. Once that the best timing to visualize the effects of treatment was estimated at 4 hours, cytotoxicity was evaluated by the Trypan Blue exclusion test: after 4 h of incubation with HU-308 or AM630 (0-1-2.5–4-5 µM), cells were collected and stained with a 0.4% Trypan Blue solution. The percentage of viable cells were counted with a TC20™ Automated Cell Counter (Biorad, Milan, Italy). Each experiment was repeated at least twice in each cell population isolated from patients.

### 4.5. Statistical Analysis

Viability data following different doses of HU-308 or AM630 were analyzed by One-Way Analysis of Variance (ANOVA), followed by Dunnett’s test for multiple comparisons to the control (untreated) condition. *p* < 0.05 was always considered as the limit for statistical significance.

### 4.6. Preparation of Samples for Stainings

Isolated cells were plated (150 cells/mm^2^ in 24-multiwells containing a glass coverslip) and allowed to attach for 48 h at 37 °C. They were then treated with HU-308 and/or AM630 2.5 µM solutions, prepared as described above (paragraph 4.3). The cells were monitored by a Leica DM IL inverted microscope after 1 h and 4 h of treatment, and then fixed, adding in each well 200 µL of 2% paraformaldehyde in PBS, pH 7.4, without washing away the treatment solution. This was done in order to allow the cells first to undergo gentle fixation, which could not alter or destroy the vesicles produced. After 10 min at room temperature, the cells underwent a second fixation for 10 min with 2% paraformaldehyde in PBS. They were then washed three times in PBS and eventually stored at 4 °C before the staining protocols described below were carried out. Each experiment was repeated at least 3 times in each cell population isolated by the patients.

### 4.7. Staining Protocols

Fixed cells were stained for 5’ in Toluidine blue 1% solution at room temperature and then rinsed 3 times in PBS.

The second staining procedure involved incubation in Alcian blue solution (pH 2.5 for hyaluronic acid, acid mucins, sulfated mucosubstances), obtained by dissolving 1 g of Alcian blue in 3% acetic acid 100 mL (glacial acetic acid 3 mL, H_2_O 97 mL). After 1 h of incubation at room temperature, cells were rinsed with 0.1 M HCl (solvent of stain) and then washed twice in 1X PBS solution. The mucosubstances appeared stained blue.

### 4.8. Immunocytochemistry to Detect HABP (Hyaluronic Acid Binding Protein)

Endogen peroxidases were blocked with 0.5% H_2_O_2_ in PBS for 10 min at room temperature. After repeated washes in PBS, the endogen biotin was blocked with ready-to-use reagents (Biotin-Blocking System, Dako, Carpinteria, CA, USA): samples were incubated for 10’ with avidin 0.1% solution, washed in distilled water, and incubated for 10’ with biotin 0.01% solution. After repeated washing in distilled water, samples were then pre-incubated with a blocking buffer (0.2% bovine serum albumin, BSA, and 0.2% Triton-X, in PBS) for 60 min at room temperature. The cells were then incubated in hyaluronic acid binding protein (HABP), Bovine Nasal Cartilage, Biotinylated, (Merck Life Science S.r.l., Milano, Italy), diluted 1:1000 in the same pre-incubation buffer, and maintained overnight at 4 °C. After repeated PBS washing, cells were incubated for 30’ in peroxidase-conjugated streptavidin (Jackson ImmunoResearch Laboratories, Inc.), diluted 1:250 in the same pre-incubation buffer. The reaction was then developed with 3,3′- diaminobenzidine (Liquid DAB plus substrate Chromogen System kit; Dako) and stopped with distilled water. Negative controls were carried out by omitting- incubation with HABP, confirming the specificity of the immunostaining. Nuclei were counterstained with hematoxylin, ready to use (Dako).

### 4.9. Semithin Sections and Transmission Electron Microscopy (TEM) Analysis

Cells seeded for 48 h in 24-multiwells (150 cells/mm^2^) and treated with HU-308 2.5 µM for 4 h were fixed in 0.1 M phosphate buffered 2.5% glutaraldehyde (Serva Electrophoresis, Heidelberg, Germany) and post-fixed in 1% osmium tetroxide (OsO_4_) (Agar Scientific Elektron Technology, Stansted, UK) in 0.1 M phosphate buffer. After dehydration in a graded ethanol series, the samples were embedded in Epoxy Embedding Medium Kit (45349, Sigma-Aldrich, St. Gallen, Switzerland), and cut in semithin (0.5 µm) and ultra-thin (60 nm) sections, with an RMC Power-Tome ultramicrotome (Boeckeler Instruments, AZ). Semithin sections were stained in 1% Toluidine blue for 30 min and analyzed by light microscopy; ultra-thin sections were collected on 300-mesh copper grids and counterstained with 2% uranyl acetate and then Sato’s lead for TEM analysis.

### 4.10. Images Analyses

Images were acquired on a Leica DMR microscope (Leica Microsystems, Wetzlar, Germany; objectives 20×, 40× and 63× Leica). The ultra-thin sections were examined with a Hitachi H-300 Transmission Electron Microscope.

## 5. Conclusions

Further studies are necessary to collect quantitative data of the vesicles production by the fascial cells under specific stimuli and to improve our understanding of the functional effects of HA-rich vesicles on the matrix and target cells, as well as the duration and consequences of their effects. However, the present work may explain some links between the use of cannabinoids for medical purposes and remodeling of the fascial matrix, which leads to greater or at least temporary tissue fluidity. In addition, this work emphasizes the importance of a peripheral effect of the endocannabinoid system. Further studies may shed more light on how to produce specific drugs able to stimulate endocannabinoids receptors at peripheral level, thus creating a medical effect without drug addiction.

## Figures and Tables

**Figure 1 ijms-21-02936-f001:**
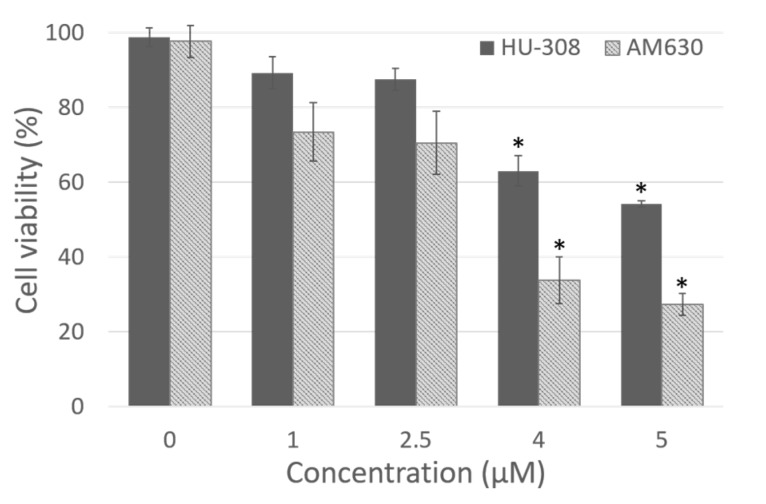
Dose dependent cytotoxicity of HU-308 and AM630 after 4 h of incubation in fibroblasts derived from human hip fascia. The control cells (0 µM) were incubated with 0.5% DMSO. *: *p* < 0.05, Dunnett’s test vs. control cells.

**Figure 2 ijms-21-02936-f002:**
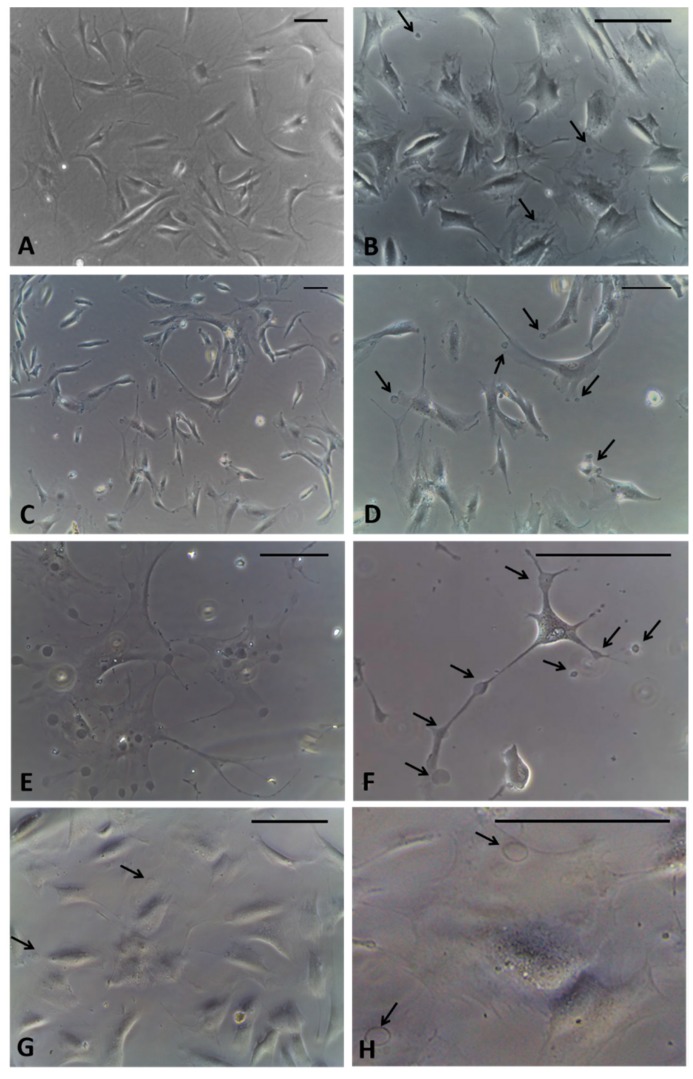
Fibroblasts derived from human hip fascia and seeded in 24-multiwells: control cells (**A**); cells treated with HU-308 2.5 µM for 1 h (**B**), 2.5 h (**C**,**D**), 4 h (**E**,**F**), and 6 h (**G**,**H**). Arrows: vesicles produced by fascial cells. Scale bars: 50 μm.

**Figure 3 ijms-21-02936-f003:**
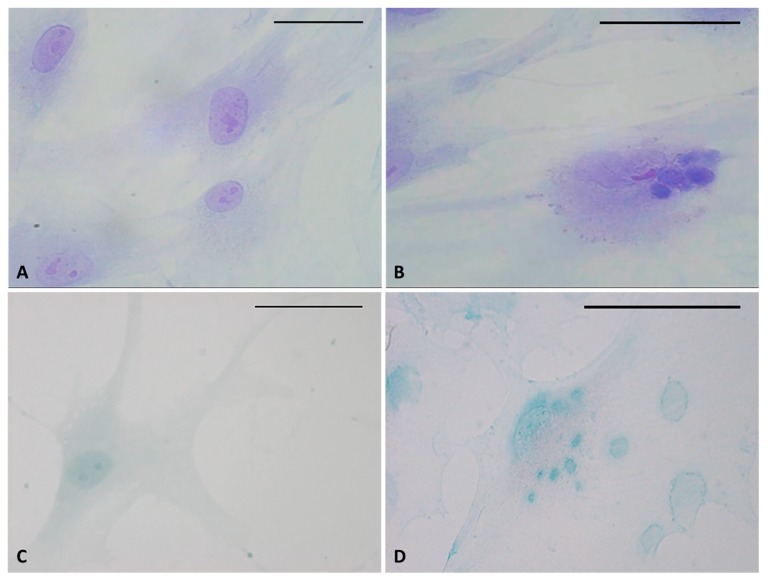
Toluidine blue (**A**,**B**) and Alcian blue staining (**C**,**D**) of fascial fibroblasts. Control cells: not incubated with HU-308 (**A**,**C**). Treated cells: incubated for 4 h with HU-308 2.5 µM (**B**,**D**). Panels B and D demonstrate stained vesicles in cell cytoplasm. Scale bars: 50 µm.

**Figure 4 ijms-21-02936-f004:**
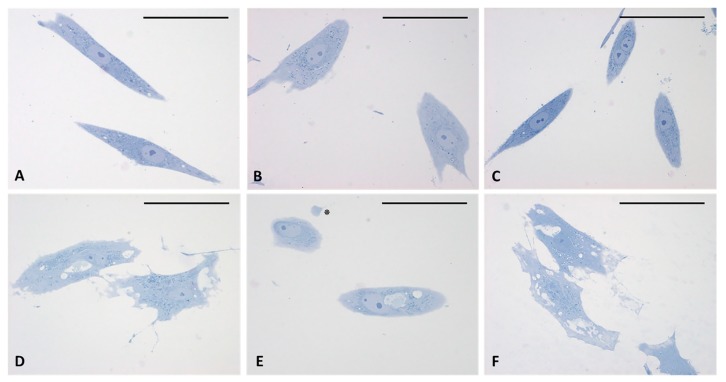
Semithin sections stained with 1% Toluidine blue. (**A**–**C**) control cells. (**D**–**F**) cells treated for 4 h with HU-308 2.5 µM. Treated cells contain vesicles and amorphous material (pale blue or white). Panel E shows a clearly exocytosed vesicle (*). Scale bars: 50 µm.

**Figure 5 ijms-21-02936-f005:**
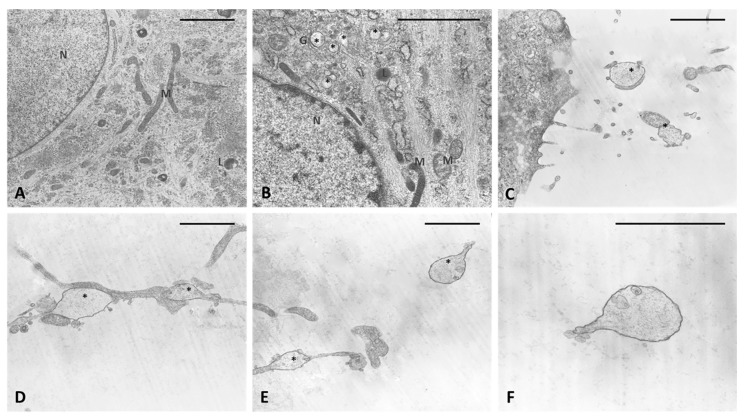
TEM analysis. (**A**) Control cells. (**B**–**F**) cells treated for 4 h with HU-308 2.5 µM. (**B**) in vesicles are visible in cytoplasm and close to Golgi complex (*). (**C**–**E**) vesicles on cytoplasmic extensions of cells or just excreted vesicles. (**F**) Detail of an exocytosed vesicle, showing amorphous material inside. Scale bars: 2 µm. N: nucleus. M: mitochondria. L: lysosome. G: Golgi complex. *: vesicles.

**Figure 6 ijms-21-02936-f006:**
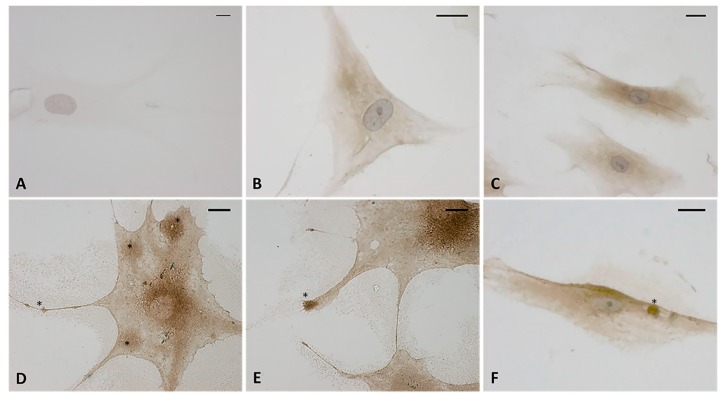
Immunostaining with hyaluronic acid binding protein (HABP) (dilution 1:1000, Merck Millipore). (**A**) Negative control with omission of HABP incubation. (**B**,**C**) control cells. (**D**–**F**) treated cells with HU-308 2.5 µM (4 h). The treated samples clearly shows HA-rich vesicles, in both the cytoplasmic region and the cytoplasmic extensions (*). Nuclei counterstained with hematoxylin. Scale bars: 10 µm.

**Figure 7 ijms-21-02936-f007:**
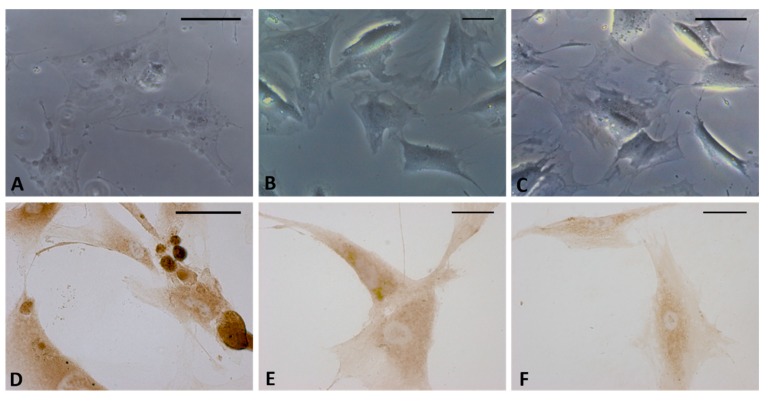
Bright field images (**A**–**C**) and immunostaining with HABP (**D**–**F**) of fascial cells treated for 4 h with HU-308 2.5 µM (**A**,**D**), AM630 2.5 µM (**B**,**E**), and HU-308 and AM630 2.5 µM (**C**,**F**). HA-rich vesicles are evident only in cells treated with agonist HU-308. Scale bars: 25 µm.

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
