# Peer review of "Sensitivity of the Fasciae to the Endocannabinoid System: Production of Hyaluronan-Rich Vesicles and Potential Peripheral Effects of Cannabinoids in Fascial Tissue"

_ijms, 2020, doi:10.3390/ijms21082936_

Round 1

Reviewer 1 Report

The authors have made a remarkable effort to improve their manuscript, which is acknowledged.

The main flaw of this manuscript is the lack of a quantitative approach to determine the production of vesicles by the cells. Since the authors did not properly quantify these vesicles, they cannot establish any conclusions on the role of CB2 on this process. Sentences such as “The number of vesicles increased with incubation time (2.5 h, Fig. 2, C-D) and, after 4 h, all cells revealed vesicles inside the cytoplasm, near the nucleus and in cellular extensions (Figure 2, E-F)” are thus not acceptable as are subjected to the personal appreciation of the researcher.

Authors should provide details on the statistical methods employed to analyze their data. For instance, fig 1 seems to reflect a t-Student analysis while 2-way ANOVA should be used. Please clarify and include a paragraph in the Materials and Methods section.

Author Response

Point 1. The authors have made a remarkable effort to improve their manuscript, which is acknowledged.

Response 1: Thank you for this comment. About the editing of English language,  all the text has been revised by a native English speaker.

Point 2. The main flaw of this manuscript is the lack of a quantitative approach to determine the production of vesicles by the cells. Since the authors did not properly quantify these vesicles, they cannot establish any conclusions on the role of CB2 on this process. Sentences such as “The number of vesicles increased with incubation time (2.5 h, Fig. 2, C-D) and, after 4 h, all cells revealed vesicles inside the cytoplasm, near the nucleus and in cellular extensions (Figure 2, E-F)” are thus not acceptable as are subjected to the personal appreciation of the researcher.

Response 2: Since we did not perform any quantification analysis, we removed all references to vesicle quantities from the text, as requested by the reviewer.

Lines 84-91: The vesicles were visible in the cytoplasm of cells even after 2.5 h (Fig. 2, C-D) and especially after 4 h, near the nucleus and in cellular extensions (Figure 2, E-F). After 6 h of incubation, the detection of vesicles visible in the cells decreased, probably because their content had already been exocytosed in the extracellular ambient (Fig. 2, G-H). Therefore, the timing of 4 hours was decided to fix the cells.

Lines 117-124: The analysis of semi-thin sections confirmed the presence of material inside the cytoplasm of the treated cells and confirmed the presence of vesicles (Figure 4, D-E-F). The nuclei appeared undamaged with respect to control cells. The latter showed no production of vesicles in any cell (Figure 4, A-B-C). TEM analysis (Figure 5) also confirmed the presence of vesicles in the treated samples, whereas cells treated only with DMSO without HU-308 appeared with no morphological changes (Figure 5, A). Cells treated with the CB2 agonist revealed vesicles near the Golgi apparatus (Figure 5, B), and also in the cytoplasmic extensions of treated cells (Figure 5, C-D-E).

Lines 163-167: We noted no differences due to the gender or age of the patient from whom we isolated the fascial fibroblasts: after only 1 h the cells seeded in the multiwells and treated with HU-308 2.5 µM started to produce vesicles, and after 4 h these  vesicles rich in hyaluronan were visible both in cytoplasm and extracellular space.

Furthermore, both in Conclusions (lines 326-329) and in Discussion sections (lines 190-193), we highlighted that further studies are necessary to collect quantitative data of the vesicles production under this specific stimulus.

Point 3:Authors should provide details on the statistical methods employed to analyze their data. For instance, fig 1 seems to reflect a t-Student analysis while 2-way ANOVA should be used. Please clarify and include a paragraph in the Materials and Methods section.

Response 2: This observation is completely correct, we apologize for the error. In this new version authors made an ANOVA test, with Dunnett's test for multiple comparison to the control (untreated) group. We obtained the same results: only doses of 4 µM and 5 µM were toxic for fascial cells, with a statistically significant decrease in viability (Dunnett’s test, p<0.05). 

These results are reported in the Legend of Figure 1 and in the Results section (lines 70-82). Furthermore we included in the Materials and Methods section the new paragraph Statistical Analysis (lines 271-274), as suggested by the Reviewer.

Reviewer 2 Report

The authors have addressed my questions and concerns appropriately. I can see the significant improvement in the revision of this manuscript. I have no more additional comments.

Author Response

Authors thank the reviewer for the positive comments about our revised and improved work. Thank you.

Round 2

Reviewer 1 Report

The interest of this manuscript is low due to the lack of a quantitative approach for the production of vesicles.

This manuscript is a resubmission of an earlier submission. The following is a list of the peer review reports and author responses from that submission.

Round 1

Reviewer 1 Report

The manuscript by Fede et al is an interesting approach to the effects of cannabinoid CB2 receptor activation in fibroblast. It seems a preliminary, though promising, study that deserves additional work:

It is necessary to make a dose-response and a time-course analysis of this experimental model. Why the dose of HU and the time of exposure were chosen is not explained or referenced.

To fully confirm the mediation of CB2 receptors, blockade with an specific antagonist is of major relevance.

Regarding methods, a more in-depth description on how images were analyzed is required. How quantifications were performed?

Please use subscripts for CB1 and CB2 throughout the text.

Reviewer 2 Report

This manuscript described the effects of the synthetic cannabinoid HU-308, a CB2R agonist, on morphology and vesicle production in cultured fascial tissues from three human subjects. It was reported that HU-308, at 2.5 uM, increased production and release of hyaluronan-rich vesicles in the cytoplama and/or extracellular matrix. Overall, it is an interesting study. However, the findings are preliminary. More studies are required to confirm this finding.

1) Only one drug dose was used. At least one more drug dose should be added to see the dose-dependent effects;

2) The authors showed only representative image data. There is no any quantitative data to show the mean group effects from multiple independent measurements;

3) The image quality of Figure 1 is poor. It is difficult to see the changes in vesicle formation and its density in those images. The densities of fibroblasts appeared to be decreased progressively from the image A to image D. A neuron-like cell is clearly shown in the image D. A specific cell marker should be used to identify the cell type presented in these images. The symbols "*" are not necessary and messed up with the "vesicles". They should be either moved or replaced by arrows. High resolution images are required to show the key finding in this study that HU-308 does increase vesicle production and release from fibroblasts.

4) to determine whether it is a CB2-mediated effect, the authors should carry out additional experiments to determine whether pre-treatment or co-administration of a CB2R antagonist (such as AM630) blocks the action produced by HU-308.

5) The authors stated that both CB1 and CB2 receptors are found in the fascial tissues. Then the authors should also consider use other cannabinoids, such as THC and WIN55,212-2 that have high affinity to CB1 and CB2 receptors, in the same study to confirm their findings with HU-308.

6) Are the vesicles the authors found in fibroblasts similar to those found in neuronal terminals? What are the major components in the vesicles of fibroblasts? What are the functional role of the vesicles released by HU-308? The authors should discuss these issues in depth in the discussion section.